# OpenReview forum: "Optimal Stopping in Latent Diffusion Models"
_ICLR.cc/2026/Conference — Submitted to ICLR 2026_

### Official Review · Reviewer_ekaL · 2025-10-22

**Soundness:** 3
**Presentation:** 3
**Contribution:** 2
**Rating:** 6
**Confidence:** 3

**Summary:**

This paper presents a theoretical analysis of an interesting observation: Early stopping in sampling may improve sample quality in latent diffusion models in a way that does not hold for ambient-space diffusion models. The authors briefly motivate their theory by demonstrating this phenomenon on a pair of latent and pixel-space models, and then introduce a toy model in which one seeks to sample from a Gaussian target distribution with diagonal covariance in a linear "latent space", and the only learnable parameters for the score function are the diagonals of the target covariance matrix. In the context of this model, the authors show that the Wasserstein distance between the model distribution and tha target distribution is not necessarily monotonic in the sampling stopping time, derive the optimal latent dimension for a given stopping time, and characterize optimal stopping times and latent dimensions for data supported on a subspace. They finally show how to generalize their results to general Gaussian targets with non-diagonal covariance matrices.

**Strengths:**

- This is generally a well-written paper, and I was able to follow the main ideas without too much trouble. The authors provide sufficient intuition for their theoretical results. There are a few instances of awkward phrasing, but these can easily be corrected in revisions. (For instance, the statement of the hypothesis "In latent diffusion models, the last diffusion steps do not improve, or even degrade, sample quality" is a bit confusing -- I initially parsed it as "the last diffusion steps neither improve nor degrade sample quality".)
- The results in this paper are correct to my knowledge.
- The main finding in this paper -- that early stopping in sampling may result in a better estimate of the target distribution when the latter is supported on a subspace -- is interesting and merits further investigation. Relaxing the Gaussian-data assumption and considering multimodal or manifold-supported distributions would be a particularly interesting future theoretical direction. However, as I explain below, I believe that the need for further empirical validation is a notable weakness of this paper.

**Weaknesses:**

My main critique of this paper is that it does not adequately demonstrate that its main hypothesis (line 50) is true in practice. The authors experiment with a single pair of latent and pixel-space models and find that early stopping in sampling improves sample quality for the latent model but not for the pixel-space model. However, it is unclear whether this is simply an artifact of the target distribution, the model architecture, or even the training procedure. (Indeed, the models are trained on different datasets, which could conceivably explain their different behavior wrt early stopping.) If this phenomenon has previously been observed in the empirical literature, the authors might include a citation. Otherwise, I'd like to see a bit more evidence that this phenomenon actually exists. Without this evidence, it is possible that the authors have developed a collection of interesting results that predict the behavior of their toy Gaussian model but that do not meaningfully bear on the behavior of real-world diffusion models.

**Questions:**

- Has the main hypothesis (line 50) been empirically validated in previous work?
- In lines 350-352, the authors write that Proposition 3 "suggests a principled guideline for practitioners: identifying and restricting the diffusion process to the data’s intrinsic dimensionality leads to more accurate and robust generative models." Is this guideline actionable in practice? I am aware of methods for estimating a data's intrinsic dimensionality using a pretrained generative model (e.g. the FLIPD estimator from Kamkari et al. 2023), but training a pixel-space model to first identify the data's intrinsic dimensionality seems prohibitively costly.

---

> ### Author Response · Authors · 2025-11-24
>
> We thank you for your feedback, we address specific questions in the following.
>
> > My main critique of this paper is that it does not adequately demonstrate that its main hypothesis (line 50) is true in practice. The authors experiment with a single pair of latent and pixel-space models and find that early stopping in sampling improves sample quality for the latent model but not for the pixel-space model. However, it is unclear whether this is simply an artifact of the target distribution, the model architecture, or even the training procedure. (Indeed, the models are trained on different datasets, which could conceivably explain their different behavior wrt early stopping.) If this phenomenon has previously been observed in the empirical literature, the authors might include a citation. Otherwise, I'd like to see a bit more evidence that this phenomenon actually exists. Without this evidence, it is possible that the authors have developed a collection of interesting results that predict the behavior of their toy Gaussian model but that do not meaningfully bear on the behavior of real-world diffusion models.
>
> >Has the main hypothesis (line 50) been empirically validated in previous work?
>
> Please refer to the general discussion above.
>
> > In lines 350-352, the authors write that Proposition 3 "suggests a principled guideline for practitioners: identifying and restricting the diffusion process to the data’s intrinsic dimensionality leads to more accurate and robust generative models." Is this guideline actionable in practice? I am aware of methods for estimating a data's intrinsic dimensionality using a pretrained generative model (e.g. the FLIPD estimator from Kamkari et al. 2023), but training a pixel-space model to first identify the data's intrinsic dimensionality seems prohibitively costly.
>
> It is a valid point. We removed the corresponding line since it is indeed misleading. However, our results still give insights on how dimensionality can have an effect on the distribution quality. In practice, people use cross-validation to pinpoint the optimal latent dimension. Note that cross-validation is a standard practice even in cases where theory suggests an optimal hyperparameter choice (e.g., for the regularization strength in ridge regression).

---

> > ### Comment · Reviewer_ekaL · 2025-11-26
> >
> > Thanks for your rebuttal. In response to my question on whether the main hypothesis has been validated in previous work, you point to the general discussion above. I assume the answer to my question is "We also note that the degradation of the FID score in the late sampling steps of latent diffusion models has already been observed in the literature (see, for example, [1] Figure 6)."
> >
> > I don't understand how this paper corroborates the existence of the early-stopping phenomenon. The discussion in Section 6.1 of [1] makes it clear that the increase in FID scores in the final sampling steps is a spurious artifact of the FID metric. In fact, the *point* of this paper is to propose a better alternative (CMMD) to the flawed FID metric. The caption to Figure 6 clearly states that the increase in FID scores in the final sampling steps is evidence of FID's pathological behavior: "CMMD monotonically improves (goes down), reflecting the improvements in the images. FID’s behavior is not consistent, it *mistakenly* suggests a decrease in quality in the last two iterations" (emphasis mine).
> >
> > This raises more doubts about the existence of the phenomenon that this paper seeks to model, which I now believe is a likely artifact of the FID metric.

---

> ### Author Response · Authors · 2025-11-27
>
> Thank you for raising your concern about the validity of the FID metric in this context. We focused on this metric given that it is overwhelmingly used in the literature, but following your comment we checked that similar conclusions hold with two alternative metrics between probability distributions, namely Sliced Wasserstein and Inception MMD (see Figure 8 in the revised manuscript). Visual inspection also confirms that the quality of the LDM-generated sample evolved marginally over the last steps when to the evolution over the last steps of the denoising process in pixel space (Figure 9). We updated the manuscript with these new experimental results.
>
> Regarding the empirical validation in previous work: thank you for giving us the opportunity to clarify our answer. To our knowledge, the observation that the quality of sample does not increase in the last steps of the latent diffusion is novel to our work. We cited the previous paper [1] as they already observe an increase in FID in the last steps of the diffusion, which is the closest observation we could find in the literature. These authors analyze this fact based on the (unsupported) assumption that the sample quality increases in the last steps, and therefore deduce that the increase in FID is an artifact of the metric. We challenge this assumption, and propose the alternative hypothesis that the sample quality decreases in the last steps. This hypothesis is backed by our experimental results with other metrics and visual inspection of the sample, as well as our theoretical results.

---

### Official Review · Reviewer_137B · 2025-10-30

**Soundness:** 3
**Presentation:** 3
**Contribution:** 2
**Rating:** 4
**Confidence:** 3

**Summary:**

The paper investigate a phenomenon in Latent Diffusion Models (LDM): the final steps in LDM can degrade sample quality more intensively compared to general diffusion models. The authors hypothesize that this phenomenon is intrinsic to the **dimensionality reduction** in LDMs. To validate this hypothesis, the authors conduct analysis in a simplified setting: Gaussian framework with linear autoencoders, quantifying the Wasserstein distance between the generated and the data distributions. The theoretical results reflect that

(1) in lower dimensional latent space, early stopping can increase the generation quality;

(2) there is an optimal latent dimension and an optimal stopping time to achieve optimal generation quality when the data has low-dim structure.

**Strengths:**

1. **Rigorous math**: the analysis is grounded with clear and clean mathematical statements reflecting the relation between dimensionality, early stopping time and sample quality in LDM.

2. **Great motivation and impact**: the theory brings a new perspective for improving the sample quality in LDM, which could also be useful in other diffusion-based tasks, such as conditional generation.

**Weaknesses:**

1. **Simplified setting**: the current setting considers only Gaussian data and linear autoencoders, which are far from practical settings. It is not clear how useful these results in this simplified setting are in practice. While some qualitative statements may extend, the quantitative statements may not. Although the paper provides numerical results on CelebA, it does not validate any of the quantitative statements in the paper directly.

2. **Split theory and experiments**: except for the degradation at the late stage, the theory and the practical numerical experiments are not very related. For example, is there a similar results to what's in Figure 4 for the LDM on CelebA?

**Questions:**

1. line 214-215: it should be replacing $\hat p_T$ by standard Gaussian;

2. line 257-260, some signs should be reflected. The current lower bound is over-conservative.

3. In Proposition 2, what does *well-ordered* mean?

4. In the training phase of the diffusion model, the score is learned by Harmonic features [1]. Is it possible to replace the dimension in the autoencoder by the features and show similar degradation phenomenon?

[1] Kadkhodaie, Zahra, et al. "Generalization in diffusion models arises from geometry-adaptive harmonic representations." arXiv preprint arXiv:2310.02557 (2023).

---

> ### Author Response · Authors · 2025-11-24
>
> We thank you for your feedback, we address specific questions in the following.
>
> >1. Simplified setting: the current setting considers only Gaussian data and linear autoencoders, which are far from practical settings. It is not clear how useful these results in this simplified setting are in practice. While some qualitative statements may extend, the quantitative statements may not. Although the paper provides numerical results on CelebA, it does not validate any of the quantitative statements in the paper directly.
>
> >2. Split theory and experiments: except for the degradation at the late stage, the theory and the practical numerical experiments are not very related. For example, is there a similar results to what's in Figure 4 for the LDM on CelebA?
>
> Please refer to the general discussion above.
>
> >1. line 214-215: it should be replacing $\hat p_T$ by standard Gaussian; 2. line 257-260, some signs should be reflected. The current lower bound is over-conservative.
>
> We thank the reviewer for pointing these out. We will change this in the next version of the work.
>
> >3. In Proposition 2, what does well-ordered mean?
>
> We meant that the sequence $t_i$ satisfies $t_1 > t_2 > … > t_n$. We will clarify this in the next version of the paper.
>
> >4. In the training phase of the diffusion model, the score is learned by Harmonic features [1]. Is it possible to replace the dimension in the autoencoder by the features and show similar degradation phenomenon?
>
> We thank the reviewer for pointing out the link between the two works. Indeed, in [2], in the comments after eq. (12), they suggest to truncate the number of representations according to the level of noise, which is also what we highlight in Proposition 2 of our work. We will discuss this connection in the next version of the paper.
>
> [2] Kadkhodaie et al., "Generalization in diffusion models arises from geometry-adaptive harmonic representations.", ICLR 2024.

---

> > ### Comment · Reviewer_137B · 2025-11-26
> >
> > I appreciate the authors' clarification, and my questions have been addressed.
> >
> > Regarding the **two weaknesses** addressed in the general comments, I have a few follow-up questions. I fully agree that *a simple setting can be valuable if it captures intrinsic behaviors of a more complex system*. However, in the current manuscript, the connection between the simplified setting (Gaussian data with linear autoencoders) and the full latent diffusion model still feels insufficiently supported. The main link demonstrated is that both systems exhibit degradation in the late stages of generation, but this alone does not fully establish that the simplified model faithfully reflects the degradation mechanism of latent diffusion.
> >
> > This is why I originally asked about reproducing experiments analogous to Figure 5 (previously Figure 4): such results would provide stronger evidence that the simple setting captures the same geometric or dynamical effects that drive degradation in latent diffusion. Even after the newly added numerical results, this type of supporting evidence still seems absent.

---

> > > ### Author Response · Authors · 2025-11-27
> > >
> > > We believe that the connections between our theory and experiments go beyond the observation on quality degradation in the last steps of the sampling procedure, and we thank the reviewer for giving us a chance to clarify this important point.
> > >
> > > Specifically, the main takeaways of our theory (encapsulated in Figure 5) are two facts: (a) the optimal latent dimension depends on the diffusion time, and for later diffusion times, a larger latent dimension is better (this leads us to predict that the FID curves for various dimensions should cross), and (b) it may happen, under certain conditions on the estimation of the score function, that the optimal sample quality is reached before the end of the diffusion process.
> > >
> > > These two facts are observable in our experiments on Imagenet, where we both observe the curves crossing and a U-shape as a function of the diffusion time (see Figures 2 and 7 in the revised manuscript). Following the reviewer’s follow-up questions, we also performed an experiment where we learn a PCA (a.k.a. a linear encoder) on MNIST, followed by learning the score with a U-Net in the latent space. In this case (Figure 10), we again observe fact (a), but not (b). This is not contradictory with our theory which tells us that (b) is not systematic, and happens when there are both a latent space with a smaller dimension and errors in score estimation. We hypothesize that MNIST being a simpler dataset, there are less errors in score estimation, which prevents from observing (b).
> > >
> > > Finally note that, in practice, the dimensions of latent spaces are usually divisors of the original dimension, because the latents are generated by convolutional neural networks. For this reason, it is not possible to generate latent spaces with arbitrary dimensions as in our synthetic experiment. This is why we can test fewer latent dimensions in the experiments with image datasets compared to the synthetic data.

---

### Official Review · Reviewer_fnHE · 2025-10-31

**Soundness:** 3
**Presentation:** 2
**Contribution:** 2
**Rating:** 2
**Confidence:** 4

**Summary:**

The paper assesses the effect of early denoising stopping on the quality of generated images in Latent Diffusion Models (LDMs). The authors theoretically analyze this phenomenon by studying diffusion processes on Gaussian mixtures and independent components, and evaluate the generated distribution using the Fréchet (Wasserstein-2) distance as a function of the stopping time. They demonstrate that, depending on the underlying data distribution and latent dimensionality, the Fréchet distance may not monotonically decrease as denoising progresses. Consequently, excessive denoising can degrade sample quality. The authors then propose two generalizations of their framework to arbitrary Gaussian mixtures and provide conditions under which early stopping is theoretically optimal.

**Strengths:**

- The paper addresses an interesting and practically relevant phenomenon—early stopping in LDMs—that has been empirically observed but rarely analyzed theoretically.
- The Gaussian analytical framework is elegant and allows for closed-form insights into the interplay between latent dimensionality, stopping time, and sample quality.
- The proofs are rigorous, and the link between early stopping and dimensionality reduction is well-motivated.
- The work provides a first theoretical grounding for empirical practice in generative modeling.

**Weaknesses:**

- The paper lacks a clear empirical conclusion linking the theoretical findings to the observed advantages of early stopping in real LDMs versus pixel-space diffusion models.
- There is a noticeable discrepancy between the theoretical setup (independent Gaussian components) and real diffusion models trained on complex datasets. The analysis is an important first step but remains far from validating the phenomenon empirically.
- The choice of Fréchet distance (FD) as the evaluation criterion is not fully justified, and it is unclear how sensitive the results are to this choice. Other metrics like KL divergence or MMD could yield different insights.
- The statement “One well-documented challenge in this method is the onset of numerical instability as the timestep t approaches 0” cites only one work and would benefit from a broader empirical basis or clearer evidence.
- Experimental validation is minimal. For a venue like ICLR, testing only one AE and one DM on CelebA-HQ is insufficient to substantiate claims about optimal stopping or to demonstrate generality across datasets or architectures.

**Questions:**

- The authors attribute the degradation at late timesteps to a low signal-to-noise ratio in the latent space.  Would this phenomenon persist in more modern architectures such as EDMs, where the model enforces unit-variance inputs and outputs at each noise level?

---

> ### Author Response · Authors · 2025-11-24
>
> We thank the reviewer for the feedback and we conducted additional experiments. Please refer to the general discussion above.
>
> >The authors attribute the degradation at late timesteps to a low signal-to-noise ratio in the latent space. Would this phenomenon persist in more modern architectures such as EDMs, where the model enforces unit-variance inputs and outputs at each noise level?
>
> We thank you for your suggestion of investigating this phenomenon using other types of diffusion models, such as EDM. We added this question as a future direction for research.

---

> > ### Comment · Reviewer_fnHE · 2025-11-26
> >
> > Thank you for your detailed rebuttal and for the additional effort in evaluating the method on higher-dimensional datasets. I appreciate these new experiments. However, I still believe that the experimental section remains somewhat limited, particularly regarding comparisons with more modern baselines such as EDM—which, despite being three years old, is still highly relevant.
> >
> > I also regret that some of the previously mentioned weaknesses were not fully addressed. In particular, the rebuttal does not provide references supporting the claim that “a well-documented challenge in this method is the onset of numerical instability as the timestep t approaches 0,” which I consider an important point to substantiate.
> >
> > I nevertheless encourage the authors to continue improving the paper, which presents solid strengths and shows promising potential. However, I would like to maintain my current score.

---

> > > ### Author Response · Authors · 2025-11-27
> > >
> > > Thank you for your thoughtful comments. We apologize for omitting to address the comment on numerical stability. We have added additional references in the revised version. Please let us know if you have any further questions.

---

### Official Review · Reviewer_81pz · 2025-11-01

**Soundness:** 3
**Presentation:** 3
**Contribution:** 3
**Rating:** 6
**Confidence:** 4

**Summary:**

The paper identifies that in Latent Diffusion Models (LDMs), continuing the diffusion process to the very end can degrade sample quality, a phenomenon not seen in standard pixel-space diffusion models. The authors hypothesize this is an intrinsic effect of the LDM's dimensionality reduction. To analyze this, they propose a simplified theoretical framework using Gaussian data and linear autoencoders. Within this setup, they analyze the Wasserstein-2 (Fréchet) distance to the target distribution, showing that lower-dimensional latent spaces benefit from earlier stopping times. The analysis is extended to consider the impact of score matching constraints and generalized from diagonal to arbitrary Gaussian covariance matrices.

**Strengths:**

* The paper highlights a specific, observable, and relevant phenomenon in LDMs: the degradation of sample quality (rising FID) in the final sampling steps. This is contrasted with pixel-space models where quality consistently improves.
* The core hypothesis—linking this degradation to the interaction between latent dimensionality and stopping time—is plausible and provides a novel perspective beyond typical arguments about numerical instability.
* The work attempts to provide a theoretical foundation for the interplay between latent dimension, stopping time, and model constraints.
* The analysis within the simplified Gaussian setting is analytically tractable and produces concrete, testable results, such as the characterization of optimal projection dimensions for given time intervals.

**Weaknesses:**

* ** Theoretical model:** The entire analysis is built on a "Gaussian framework with linear autoencoders". This is a big oversimplification. Real LDMs, like the one used in the paper's own introductory experiment (Figure 1), use highly **non-linear** autoencoders (specifically VQ-VAEs). In summary, perfect autoencoders correspond to nonlinear functions (enc and dec) such that dec(enc(x))=x - p_{0} a.s
The linear projection model $P$ does not capture the essential non-linear manifold learning performed by the autoencoder, making the relevance of the paper's entire theoretical analysis to practical LDMs questionable.

* **Technical incorrectness of the core SDEs:** The formalization of the latent diffusion process appears to be incorrect.
    1.  The paper defines the latent forward SDE as

$dP \vec{X}_{t}=-w_{t}^{2}P\vec{X_{t}}dt+\sqrt{2w_{t}^{2}}dP\vec{W_{t}}$

 for a general matrix $P \in \mathbb{R}^{d \times D}$. If $ P $ is not an orthogonal projection (i.e., if $ PP^\top \neq I_d$), then $P\vec{W}_t$ is **not** a standard d-dimensional Brownian motion (I_d covariance). This defines a different diffusion, and its time-reversal is more complex than stated.

 The backward SDE in Equation (2) is (**when P is not an orthogonal projection**) wrong. As shown in Anderson (1982, eq. 3.12), the correct reverse drift involves metric tensor terms ($g^{ij}$) induced by the projection, which are missing from Equation (2).

* **Misleading justification:** The paper claims its simplified setting "already exhibits phenomena similar to the larger-scale evidence", but this is just an assertion. The link between the non-monotonic FID in Figure 1 (a complex, non-linear LDM on CelebA) and the non-monotonic Fréchet distance derived from a misformulated linear-Gaussian model (Proposition 1) is never established. The "phenomenon" may be similar, but the causes are possibly distinct.
* **Esperimental validation**. As the considered distributions are particularly simple (dimensionality and structure) the authors could have included some validation of the results with a *real* score network and compare the results

**Questions:**

*  How can the theoretical results, derived from a simplified linear-Gaussian model, be claimed to explain a phenomenon observed in a highly non-linear VQ-VAE-based LDM? What evidence supports the claim that the *causes* are the same, rather than just the high-level behavior (non-monotonicity)?
*  The backward SDE in Equation (2) appears to be valid *only if* $PP^\top = I_d$. However, the setup in Equation (1) is for a general $P \in \mathbb{R}^{d \times D}$, which does not guarantee this. Can the authors clarify on this in the text?

*  In Proposition 1, the condition for non-monotonicity depends entirely on the *estimation error* ($\sigma_{d'}$ vs $\hat{\sigma}_{d'}$). This suggests the phenomenon is an artifact of estimation. However, the paper's introduction claims the phenomenon is "intrinsic to the dimensionality reduction", *not* estimation error. How do you reconcile this? The proof for the *true* score (first part of Prop. 1) is shown to be monotonic.

* The notation d_F(...) =min_{} d_F is highly confusing. Can the authors clarify in the text?

* Minor: why just Frechet Distance? Being the distributions Gaussian, other divergences (e.g. KL) are available. Can the authors present some results on that as well?

---

> ### Author Response · Authors · 2025-11-24
>
> We thank you for your feedback, we address specific questions in the following.
>
> W1, W3, W4, Q1: Please refer to the general discussion above.
>
> >The backward SDE in Equation (2) appears to be valid only if $PP^\top=I_d$. However, the setup in Equation (1) is for a general $P\in\mathbb{R}^{d\times D}$, which does not guarantee this. Can the authors clarify on this in the text?
>
> Thank you for your remark. We have indeed omitted when introducing the notation to specify that the projection matrix should be orthogonal, and in this case the backward SDE is correct. This does not affect the rest of our work, as we only consider orthogonal projections and our results are correct. We are now clearly stating this assumption when introducing the notation to clarify this point.
>
> >In Proposition 1, the condition for non-monotonicity depends entirely on the estimation error. This suggests the phenomenon is an artifact of estimation. However, the paper's introduction claims the phenomenon is "intrinsic to the dimensionality reduction", not estimation error. How do you reconcile this? The proof for the true score (first part of Prop. 1) is shown to be monotonic.
>
> Thank you for this very sharp observation. What our results show is that non-monotonicity occurs when estimation error and dimension reduction are present. We phrased it in this manner because in practice, estimation error is inevitable and always present, but we will emphasize this point in the next version of the paper.
>
> >The notation d_F(...) =min_{} d_F is highly confusing. Can the authors clarify in the text?
>
> We thank the reviewer for pointing this out. We will modify accordingly in the next version of the work.
>
> > Minor: why just Frechet Distance? Being the distributions Gaussian, other divergences (e.g. KL) are available. Can the authors present some results on that as well?
>
> Please refer to the general discussion above.

---

### Author Response · Authors · 2025-11-24
**Thanks to the reviewers for the reviews**

Dear reviewers,
We warmly thank you for your time and relevant comments, which will help us improve our work. If accepted, we intend to take into account your suggestions. We answer the specifics of questions pointed out by the reviewers in individual responses.

Sincerely,

The authors



**Additional experiments.** We have addressed a common concern across reviewers that there is still a gap between our theoretical work and empirical observations. To address this, we include new experiments on ImageNet to compare the FID of diffusion models trained on pixel space and latent space. We also note that the degradation of the FID score in the late sampling steps of latent diffusion models has already been observed in the literature (see, for example, [1] Figure 6). We have conducted additional numerical experiments on the dataset ImageNet, for which two latent diffusion models were trained, respectively in dimensions $32\times 32\times 4$ (LDM-8) and $64\times64\times3$ (LDM-4). In this experiment, beyond the simulated-data setting, we observe the same phenomenon: for both latent models, the best generations (according to the FID score) occur earlier in the backward diffusion process. This provides additional support for our finding: there is a need to early-stop the sampling phase. These results are consistent with our theoretical findings in a simplified setting involving Gaussian distributions.

**Analysis in a Gaussian setting.**  Considering a simplified setting, for the theoretical analysis of a very complex system, is a common scientific method, which allows us to isolate and understand fundamental mechanisms within it. We actually consider it a strength of our analysis that we are able to exhibit this behavior across complexities, from trained neural auto-encoders and score functions on real data, to our Gaussian setting. This strongly suggests that the root of this phenomenon can indeed be understood through this lens. To further strengthen this claim, we have nevertheless added additional experiments, as described above.

**Choice of Fréchet Distance.** This distance is the de facto standard metric to evaluate generative models, with a focus on perceptual aspects of the images. It is not possible in our work to use the KL divergence because the distributions of interests are degenerate. We expect the Maximum Mean Discrepancy (MMD) to yield similar results, and will investigate this in the future.

[1] Jayasumana, Sadeep, et al. "Rethinking fid: Towards a better evaluation metric for image generation." Proceedings of the IEEE/CVF Conference on Computer Vision and Pattern Recognition. 2024.

---

### Author Response · Authors · 2025-12-02
**Summary for AC**

Dear AC,

Given the exceptional circumstances regarding the ICLR review process, we kindly summarize the discussion with the reviewers preceding the end of the rebuttal.

The paper studies a phenomenon specific to latent diffusion models (LDMs): sample quality can decrease in the final denoising steps. We propose a theoretical explanation grounded in a Gaussian–linear latent-space model, showing that the interaction between latent dimensionality and score estimation error can produce a non-monotonic distance to the target distribution as a function of the stopping time. Reviewers agree that the paper is mathematically rigorous, clearly written, and that the phenomenon is interesting and relevant for generative modeling.

The main criticisms concern (i) the simplified linear–Gaussian setting, (ii) the need for stronger empirical validation, and (iii) the role and limitations of FID as an evaluation metric. While we emphasize that considering a simplified setting allows us to isolate and understand fundamental mechanisms within it, we also expanded experiments substantially to address the reviewers’ comments. New ImageNet and MNIST experiments show our two theoretical predictions—curve crossings across latent dimensions and U-shaped behavior in stopping time—beyond the original CelebA-HQ experiment. We further verified that the degradation is not an artifact of FID by reproducing it with Sliced Wasserstein and Inception MMD, and by visually inspecting the images to verify that early stopping the latent diffusion leads to cleaner images then when early stopping the pixel diffusion. Reviewers’ other questions about notation, assumptions, and qualitative claims were taken into account.

We thank the reviewers for the thorough discussion, which allowed us to significantly improve the paper.

Best,

The authors

---

### Meta-Review · Area_Chair_LSbH · 2025-12-14

**Summary:**

This paper studies a phenomenon in latent diffusion models: the final denoising steps can degrade sample quality, unlike typical pixel-space diffusion. The authors attribute this to dimensionality reduction in LDMs and analyze it in a simplified Gaussian setting with linear autoencoders, evaluating mismatch via the Wasserstein-2 distance as a function of stopping time and latent dimension. The theory shows (i) early stopping can improve quality in lower-dimensional latent spaces, and (ii) when data has low-dimensional structure, there exist an optimal latent dimension and an optimal stopping time.

However, the evidence is largely grounded in a simplified theoretical setting; while the paper includes additional ImageNet experiments and other supportive evaluations (e.g., using metrics beyond FID), reviewers generally found the empirical support still insufficient, and the manuscript in its current form does not yet meet the bar for publication.

**Reviewer Concerns:**

Reviewers converge on two core concerns. First, the theoretical analysis is viewed as overly simplified (raised by Reviewers 81pz and 137B), limiting how strongly it can substantiate the proposed explanation for the phenomenon in LDMs. Second, several reviewers (fnHE, 137B, ekaL) find the empirical evidence insufficient to support the paper’s main claim that “in latent diffusion models, the last diffusion steps do not improve, or even degrade, sample quality”, and worry the current experiments do not sufficiently demonstrate that this holds in practice.

**Reviewer Scores:**

* Reviewer 81pz: The potential missing condition that (P) is orthogonal is addressed. However, the root-cause explanation / the gap between Figure 1 and Proposition 1 does not appear fully resolved. Score likely unchanged.

* Reviewer fnHE: The request for comparisons against more modern diffusion baselines is not fully addressed. Score likely unchanged.

* Reviewer 137B: The connection between the simplified theory setting (Gaussian data with linear autoencoders) and full latent diffusion models remains insufficiently supported in the current manuscript. Score likely unchanged.

* Reviewer ekaL: The concern that the claim around Line 50 is not fully addressed and that FID alone is insufficient remains. While the authors added MMD and Sliced Wasserstein, reviewers still expect a more careful and thorough investigation. Score likely unchanged.

---

### Decision · Program_Chairs · 2026-01-26

Reject